# Protocols of Thiamine Supplementation: Clinically Driven Rationality vs. Biological Commonsense

**DOI:** 10.3390/jcm14113787

**Published:** 2025-05-28

**Authors:** Mickael Bonnan

**Affiliations:** Neurology Department, Hôpital Henri-Mondor, 1 Rue Gustave Eiffel, 94000 Créteil, France; mickael.bonnan@aphp.fr; Tel.: +33-(0)1-49-81-23-04

**Keywords:** thiamine/administration and dosage, thiamine deficiency, Wernicke encephalopathy, beriberi

## Abstract

**Background:** Thiamine deficiency requires supplementation to prevent or reverse severe clinical complications. Are supplementation protocols correctly tailored in neurology? **Methods:** An examination of recommended thiamine supplementation protocols is undertaken. **Results:** Recommended thiamine levels are much higher than required, reaching a median daily intake and total amount of 7.8-fold and 30.6-fold greater than the whole-body thiamine store, respectively. The high initial intake is often followed by tapering over days or weeks. **Conclusions:** Thiamine supplementation protocols in neurology mostly recommend far higher doses than those biologically required and could probably be simplified to a single 100 mg dose injected as early as possible.

## 1. Introduction

Vitamin deficiencies in neurology are associated with well-defined clinical patterns that are familiar to medical students and which are easily prevented or treated. Vitamins are organic compounds that cannot be synthesized by the body and which are essential to its survival. Their availability in the body mainly depends on three parameters: mean daily intake, whole-body store, and metabolic loss. One may assume that the mean recommended intake of vitamins is intended to compensate for their loss. Setting aside clinical issues associated with an increased metabolic loss of vitamins, vitamin deficiencies may be considered as the result of chronic lower-than-required vitamin intake leading to a low whole-body store. Replenishment alone should fill the gap between actual and normal vitamin reserves in association with modified daily intake to prevent early relapse. While a slight excess of vitamin supplementation may temporarily be required to correct deficient metabolic pathways, the correction of vitamin deficiencies from a biological point of view would require a vitamin intake that is roughly equal to the normal whole-body store.

Thiamine is probably one of the most frequently supplemented vitamins. However, do recommended supplementation protocols really achieve the goal of ensuring this biological balance? To address this issue, we assessed thiamine levels in pills and other regimens commonly proposed to correct thiamine deficiency.

## 2. Materials and Methods

Thiamine supplementation products and drugs containing thiamine available in France were analyzed for the following: their whole-body level recommended daily intake and bioavailability of different routes. Recommended treatment regimens, which slightly vary from one manual to another, were sampled from a wide range of sources. The recommended dosage, route, and duration of treatment were obtained and adjusted for route bioavailability. Extreme dosage ranges proposed by the manuals were always collected, and the lowest and highest bounds of dosages were estimated as the total amount and mean daily amount. Each treatment regimen generated four variables: the mean daily intake and the total amount of absorbed vitamins, which were both corrected for the lowest bioavailability of oral intake (formulae are available in Appendix A). These values were then compared to the highest known values of the whole-body store. Values were given as the median [IQR 25;75%]. Statistical analysis was performed with R© (v.4.4.2). All generated data are available in Appendix A.

## 3. Results

### 3.1. Collected Data

Fourteen papers provided 59 different regimens for preventive or curative supplementation.

### 3.2. Bioavailability of Thiamine and Whole-Body Store

The average daily requirement is 0.33 mg per 4400 kJ of energy, or 1 to 1.5 mg per day in adults, and the whole body store ranges from 20 to 30 mg [1,2]. Intestinal thiamine absorption depends on complex regulation mechanisms [3], and the bioavailability of oral thiamine is a matter of debate. Earlier work suggested that the maximum amount of absorbed thiamine in healthy subjects was 4.8 mg after a single 20 mg oral dose and only 1.5 mg in alcoholic patients [4,5], whereas the transportation rate can be saturated at doses higher than 20 mg to 50 mg [6]. Isolated cases of patients failing to improve despite receiving oral thiamine are supportive of these findings [7,8]. However, more recent papers do not confirm this limited absorption rate. Thiamine transportation from the intestine could follow both an active absorption mechanism at low concentrations and passive mechanisms at high concentrations, which were found to be linear and not saturable up to 1500 mg [1,9], and similar absorption was achieved [10]. The clearance of blood thiamine is obtained by kidney filtration. About 5% is recovered in urine after oral administration, whereas about half is recovered after IV injection [11]. To justify prescribing high doses over a long period, several authors have used the values estimated by Thompson et al., who proposed that oral bioavailability could be saturated to 4.5 mg after each oral intake [12]. Although this hypothesis has recently been challenged, this low bioavailability rate has been used for adjustments to minimize the calculated amount of absorbed thiamine (corrected values).

### 3.3. Amounts of Thiamine in Available Brands

Thiamine is available in France in 16 products sold by different companies for oral (n = 9), IV (n = 6) or IM (n = 1) administration. First, we noted the thiamine levels in the pills or vials. Single-dose levels varied greatly from 1 to 500 mg, with a median value that was more than twice as high as the whole-body store (74 mg [8.4; 250]). All the products containing a lower amount than the whole-body store were designed for preventive supplementation purposes together with other vitamins or multi-drugs in which thiamine plays an additional role (e.g., Hexaquine^®^ contains 32 mg of thiamine). These later products, mainly dedicated to nutrition (3 IV, 4 PO), were discarded from further analysis (Nutrients in Appendix A). Concerning brands designed to cure deficiencies, the minimal amount of thiamine is 250 mg in oral pills and 100 mg in IV vials, with the highest doses being 250 mg and 500 mg, respectively (Figure 1A). Therefore, these oral and IV products contain 3.3-fold to 16.7-fold more thiamine than the whole-body store, respectively.

### 3.4. Supplementation Regimens Deliver Very High Amounts

All the regimens except six (10.3%) contained unitary doses higher than the whole-body store. Among the exceptions, three contained 20 mg per intake (the lower bound of the estimated whole-body store), and the others contained 5–10 mg. Subsequent calculations were made using the more conservative values in the literature that are used by many authors, even though they minimize the bioavailability of products taken per os. Less conservative estimates (uncorrected values) would provide far higher thiamine than whole-body store values, whatever the regimen (Figure 1B). Among the regimens, daily intake and whole doses reached median amounts of 7.8-fold and 30.6-fold for the whole-body store of thiamine, respectively (Figure 1C). Outlier values of amounts equal to or less than the whole-body store were all low-dose oral supplementations given in trials or recommended to treat minor polyneuritis.

Supplementation was intended both for preventive and curative purposes. Curative regimens had a higher dose than the whole-body store in 90% of mean daily doses and in all cumulated doses, whereas 50% and 94% of the mean daily and cumulated doses, respectively, in preventive regimens contained a dose exceeding the whole-body store, although the amounts were often lower than in curative regimens.

### 3.5. Clinical Expectation Is Reflected in Biological Supplementation

Thiamine deficiency may induce *wet beriberi* due to cardiac insufficiency or *dry beriberi*, which is associated with neuropathy and Wernicke–Korsakoff syndrome. Subclinical patterns are observed mostly in alcoholic patients who often lack thiamine due to its decreased absorption when associated with alcohol. Decompensation may suddenly occur in alcoholics during hospitalization. In neurology, the clinical patterns are sometimes severe owing to extensive tissue destruction: axonal lesions in neuropathy and the loss of neurons in Wernicke–Korsakoff syndrome. Consequently, the delay in clinical improvement is proportionate to the severity of the clinical pattern at onset. On the other hand, the clinical monitoring of minor impairment often demonstrates the alleviation of symptoms within hours after the correction of thiamine deficiency, with the time to resolution of confusion, ataxia, and ocular anomalies ranging from 1 to 3.5 days [13].

Most of the thiamine supplementation regimens analyzed were based on clinical expectation rather than on the expected correction of deficiencies, and 95% of protocols were explicitly associated with a specific clinical context ranging from the suspicion of thiamine deficiency in alcohol-dependent patients to severe Wernicke syndrome. The route proposed was IV/IM in 80% of patients and was tapered to oral in 25% of patients. The maximal duration of treatment was either not mentioned or was based on clinical improvement in 32% and 8% of the regimens, respectively. An extreme consequence would be to treat severe Wernicke patients until they improve, which is a goal that could take months or even never be attained.

## 4. Discussion

All the regimens analyzed provide more than enough thiamine. In most cases, a single injection sufficed to resolve a complete whole-body store depletion. In all cases, appropriate supplementation cured the deficiency, followed by an adequate maintenance dose to prevent relapse, i.e., 2 mg daily. Once the whole-body store was fully restored, the patient improved clinically, so a longer duration of supplementation was unlikely to provide better improvement [12].

Apart from clinical monitoring, there is no biological monitoring technique to confirm that thiamine-dependent metabolic pathways and stores are replenished. On the other hand, *shoshin beriberi*, which involves cardiogenic shock leading to death, is associated with lactate acidosis. Thiamine is a cofactor of pyruvate dehydrogenase (PDH), which converts pyruvate to acetyl-CoA in aerobic metabolism at the beginning of the Krebs cycle [14]. In thiamine deficiency, this pathway is diverted to an anaerobic metabolism via lactate dehydrogenase (LDH), which converts pyruvate to lactate. For this reason, the injection of glucose prior to the correction of thiamine deficiency increases lactate production and leads to lactic acidosis, so it should be avoided. The lactate level is commonly normalized within hours after injection of 50 mg to 100 mg of thiamine, and cardiac signs are completely reversed within hours or days [15,16,17,18,19,20]. These examples strongly support the hypothesis that low-dose thiamine is sufficient to reverse the deficiency, even though the low dose is still higher than the whole-body store. Lactic acidosis is involved in Wernicke syndrome since the lactate level is high in the cerebrospinal fluid and on grey lesion spectroscopy [21], and astrocytic cells concentrate lactate and drive focal brain edema [22]. Unfortunately, the kinetics of lactate reversal in the brain cannot be easily monitored after thiamine supplementation.

Most of the regimens analyzed include specifications derived from other medical disciplines. The tapering of drugs is a common practice aimed at preventing drug withdrawal (e.g., benzodiazepine) or at determining the minimal level required to control the disease (e.g., steroids). The clinician closely monitors the patient’s clinical signs to determine whether the reduction in dose allows the symptoms to relapse so tapering is adjusted to the symptoms and the disease. Thiamine was also tapered in 30% of the cases investigated, at least from the initial parenteral route to the oral route. The whole infused dose of thiamine was not considered an objective to be reached by the end of treatment. Indeed, none of the manuals made this calculation, and the whole dose was never compared with the whole-body store, so the recommendation was always massive administration.

Fortunately, setting the value far higher than necessary has almost no consequence, given the very low cost of thiamine and the absence of toxic side effects. On the other hand, there is a simple take-home message about thiamine supplementation: administer the first dose as early as possible, whatever it is, given that the lowest available dose contains more thiamine than the whole-body store.

A retrospective series of patients with *shoshin beriberi* failed to demonstrate any effect of the initial dose (median 100 mg/d [60; 300]) [19,23]. Concerning the prevention of cognitive impairment, randomized trials using lower amounts of thiamine ranging from 5 to 200 mg demonstrated a maximal benefit at higher doses [24], although the results should be carefully examined. Patients receiving a single 100 mg dose had the same outcome as those receiving a higher dose [13], while doses ≥ 500 mg/day did not lead to a better outcome [23]. Another trial, testing different daily doses from 100 to 1500 mg to prevent Wernicke syndrome, failed to demonstrate any inter-group difference [25]. These results suggest that unless the dose is minimal, e.g., 5 mg per day, the intake can be so massive that it prevents any dose–effect relationship being observed. Altogether, the evidence points to the efficacy of a single IV or IM 100 mg dose. If administration is per os, the same dose could be given every day for three days, given the putative lower bioavailability of the oral route. Although the optimal dose still requires validation by controlled trials, complex patterns of supplementation seem to be excessive. On the other hand, supplementation should be given as early as possible.

The question arises of why the obvious has traditionally taken a back seat in favour of the over-complex. Clinicians may be faced with vitamin-deficient patients expressing more or less severe signs that persist over time, owing to the specific kinetics of remission. In severely affected patients, it is tempting to scale up the treatment in an attempt to counter the persistence of symptoms, in the belief that higher amounts mean greater benefit. Cerebellar symptoms and Wernicke syndrome are often cured within hours, and supplementation may be used as a diagnostic test. Landmark work published by Victor, Adams, and Collins in 1971 observed that the earlier improvements occurred within hours of administering 50 to 100 mg of thiamine orally [26]. On the other hand, painful neuropathy and Korsakoff syndrome tend to persist for weeks or months after an acute onset. They cannot be improved by higher doses or a longer period of administration but rather require time for axonal sprouting to take place. Although the use of massive thiamine doses over several days is often put forward to explain the delay in improvement (e.g., in the patient from [27]), the latter is merely due to the kinetics of neurological vicariance rather than to cumulated doses (Figure 2).

The single 100 mg dose of thiamine now seems to be widely used [28], although clinical trials should be designed to compare the efficacy of reduced and traditional doses. In the event of an intestinal block, an IV/IM injection is required [6]. An IV push of thiamine has rarely been associated with anaphylaxis, and tolerance was good at doses below 200 mg [29,30,31]. It is preferable to dilute IV thiamine (e.g., 50 cc saline over 10 min) or to use an IM injection.

## 5. Conclusions

Patterns of supplementation for thiamine deficiency are unnecessarily complex. A single dose given as early as possible in the window of opportunity is sufficient to reverse “biochemical lesions” and prevent permanent damage [6,26]. Intake in randomized controlled trials is so high that it cannot demonstrate any dose–effect relationship.

## Figures and Tables

**Figure 1 jcm-14-03787-f001:**
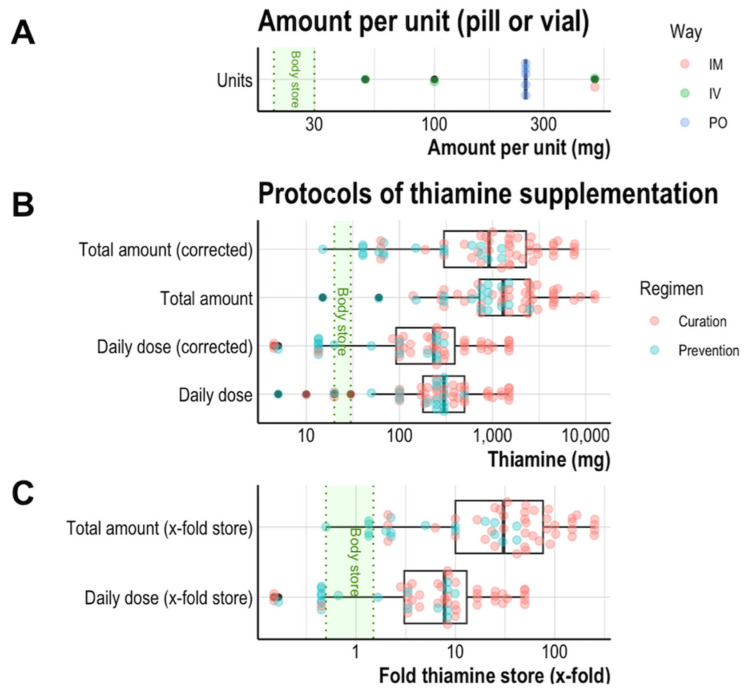
Substitution of thiamine deficiency. Data are plotted showing the admitted range for the normal whole-body thiamine store (20 to 30 mg, green area). (**A**) The amount of thiamine in commercially available pills. (**B**) The mean daily dose and total amount defined as the cumulation of each daily dose over the complete regimen. Corrected amounts are adjusted for the lowest estimated oral bioavailability. (**C**) The same values plotted as folds of the upper limit of the whole-body thiamine store. For example, this means that the median daily thiamine dose provided by supplementation protocols is approximately 10 times higher than the upper limit of the estimated whole-body thiamine store. The results are expressed as median and interquartile range.

**Figure 2 jcm-14-03787-f002:**
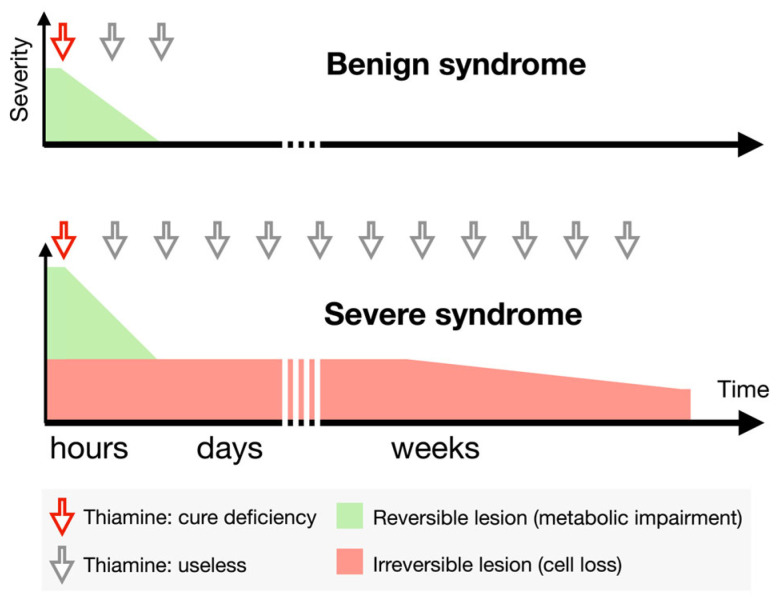
Kinetics of clinical improvement after thiamine supplementation. Benign patients mostly suffer from metabolic impairment (or “biochemical lesions” [26], e.g., lactic acidosis and minor Wernicke syndrome), which can be reversed within hours or days after thiamine injection, which cures deficiency (red arrows). In the most severe patients, metabolic impairment may induce cell loss and irreversible lesions (e.g., neuropathy and Korsakoff). Improvement is expected to follow a slow kinetic and to remain incomplete. Discrepancies between the kinetics of improvement of the functional and destructive lesions lead to unnecessary sustained thiamine supplementation (grey arrows).

## Data Availability

No new data were created. All data are provided in the Appendix A.

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
