# Peer review of "Protocols of Thiamine Supplementation: Clinically Driven Rationality vs. Biological Commonsense"

_jcm, 2025, doi:10.3390/jcm14113787_

Round 1
Reviewer 1 Report
Comments and Suggestions for Authors
I commend the author for trying to bring rationality and clear thinking to the dosing of what is fundamentally a nutrient and not a pharmaceutical in a clinical setting. By analogy, I have often wondered why clinical neurologists dutifully prescribe riboflavin at 400 mg/day for migraine prophylaxis when this dose has merely been borrowed from studies of frank mitochondrial Complex II deficiencies.
Adding to the author’s point, Tsepkova, et al. (2017), report that a single, extremely high dose of thiamine in rats in some cases can cause a compensatory downregulation of the activity of the associated enzyme (e.g., alpha-ketoglutarate dehydrogenase), mimicking a deficiency.
Having said this, I think the argument based on whole-body stores is simplistic as it seems to imply a rapidly attained tissue saturation of free-form thiamine. I would think that most of the tissue stores are of thiamine bound to its associated enzyme complexes and these complexes take time to construct. (In contrast, the blood half-life of thiamine is about 96 minutes; Tallaksen, et al., 1993.) The “whole body store” standard seems contradicted by the Ambrose, et al. study (reference 21) in which IM thiamine was given at doses of 5, 10, 20, 50, 100, or 200 mg a day for two days: “A planned comparison between the 200 mg group and the mean of the other dosage groups was significant…” (Ambrose et al., 2001, p. 114). As whole body stores are 20-30 mg, all doses above 20 mg should have been equivalent. Also, the ability to use and store thiamine likely depends on the Michaelis dissociation constant for the associated enzymes, which are certainly altered in enzyme polymorphisms, mutations (Ames, et al., 2002) and aging (Ames, et al., 2006) and might be altered in disease states such as Wernicke’s encephalopathy (Galvin, et al., 2010).
Nonetheless, the author makes a good case when reviewing data for the rapid resolution of lactic acidosis and cardiac symptoms and the relatively flat dose-response curve above I00 mg IM. I think emphasizing this type of data and providing a more nuanced understanding of thiamine stores will strengthen an already good manuscript.
Ames BN, Elson-Schwab I, Silver EA. High-dose vitamin therapy stimulates variant enzymes with decreased coenzyme binding affinity (increased K(m)): relevance to genetic disease and polymorphisms. Am J Clin Nutr. 2002 Apr;75(4):616-58. doi: 10.1093/ajcn/75.4.616. PMID: 11916749.
Ames BN, Suh JH, Liu J (2006) Enzymes lose binding affinity (increase Km) for coenzymes and substrates with age: A strategy for remediation. Nutrigenomics: Discovering the Path to Personalized Nutrition, eds Kaput J, Rodriguez R (John Wiley & Sons, Hoboken, NJ), pp 277–293.
Day E, Bentham P, Callaghan R, Kuruvilla T, George S. Thiamine for Wernicke‐Korsakoff Syndrome in people at risk from alcohol abuse. Cochrane Database of Systematic Reviews 2004, Issue 1. Art. No.: CD004033. DOI: 10.1002/14651858.CD004033.pub2. Accessed 06 May 2025.
Galvin R, Bråthen G, Ivashynka A, Hillbom M, Tanasescu R, Leone MA; EFNS. EFNS guidelines for diagnosis, therapy and prevention of Wernicke encephalopathy. Eur J Neurol. 2010 Dec;17(12):1408-18. doi: 10.1111/j.1468-1331.2010.03153.x. PMID: 20642790.
Tallaksen C, Sande A, Bøhmer T, Bell H, Karlsen J. Kinetics of thiamin and thiamin phosphate esters in human blood, plasma and urine after 50 mg intravenously or orally. Eur J Clin Pharmacol 1993; 44: 73–78.
Tsepkova, P.M.; Artiukhov, A.V.; Boyko, A.I.; Aleshin, V.A.; Mkrtchyan, G.V.; Zvyagintseva, M.A.; Ryabov, S.I.; Ksenofontov, A.L.; Baratova, L.A.; Graf, A.V.; et al. Thiamine induces long-term changes in amino acid profiles and activities of 2-oxoglutarate and 2-oxoadipate dehydrogenases in rat brain. Biochemistry 2017, 82, 723–736.
Author Response
Comment 1: I commend the author for trying to bring rationality and clear thinking to the dosing of what is fundamentally a nutrient and not a pharmaceutical in a clinical setting. By analogy, I have often wondered why clinical neurologists dutifully prescribe riboflavin at 400 mg/day for migraine prophylaxis when this dose has merely been borrowed from studies of frank mitochondrial Complex II deficiencies.
Response 1: We understand the reviewer position and we partially agree that doses are sharply different between nutrients and pharmaceutical drugs. We discarded nutrients from this analysis.
Fig 1A now shows more clearly how single doses are already high.
We are also puzzled by the use of riboflavin in migraine: French neurologists use to say that it remains a specific German tradition. I have no personal idea about the real efficiency of this drug.
Comment 2: Adding to the author’s point, Tsepkova, et al. (2017), report that a single, extremely high dose of thiamine in rats in some cases can cause a compensatory downregulation of the activity of the associated enzyme (e.g., alpha-ketoglutarate dehydrogenase), mimicking a deficiency.
Response 2: We read the article which is highly challenging. I'm not really sure about the described effects of thiamine. Moreover, huge doses were used, which are even far higher (400mg/kg) than those used in human medicine and might provoke different effects. We chose not to add this reference.
Comment 3: Having said this, I think the argument based on whole-body stores is simplistic as it seems to imply a rapidly attained tissue saturation of free-form thiamine. I would think that most of the tissue stores are of thiamine bound to its associated enzyme complexes and these complexes take time to construct. (In contrast, the blood half-life of thiamine is about 96 minutes; Tallaksen, et al., 1993.).
Response 3: We agree that the concept of whole-body store remains an oversimplification of a complex biochemical landscape implying protein binding sites, differential Km, various transmembrane transporters, and so on. Whatever the mechanisms underlying the thiamine store, a kind of whole-body store does exist since it could be depleted under an appropriate thiamine-depleted diet, and then replenished. As a consequence, it seems correct to approximate that a certain amount of thiamine is available in the body (literature admits 20 to 30mg, but a thrice higher amount would not dramatically change our results): average daily requirement is about 1-1.5mg/day, which is in agreement with a 3 weeks thiamine-deficient diet required to trigger signs of deficiency.
The whole-body store does not preclude any kind of metabolism, or storage form. We agree that associated enzyme complexes, if required, would probably take time to construct. However, as Tallaksen pointed out: urinary excretion is very low (about 2.5% of the ingested dose). Metabolism is probably very quick, since serum monophosphate thiamine (TP) and diphosphate (TPP) reach their peak within 2h post-ingestion, suggesting that cell transport is probably largely involved in the blood clearance.
Tallaksen suggested that most of the ingested thiamine was not absorbed. However, Smithline, 2012 and Weber 1985 did not confirm this limited absorption rate, and rather observed a linear absorption rate, non-saturable up to high concentration. We added the reference to Baines (PMID: 3358822) who showed that same levels of thiamine diphosphate were reached after parenteral or PO doses.
That's why we always calculated 2 ingested doses: one is the sum of all the ingested thiamine, the other is corrected to ceil ingested thiamine to 4.5mg (maximum amount of absorbed thiamine in heathy subjects: Thomson 1970, 1972 and 2002) whatever the ingested dose. We assume this sharply conservative ceil of absorbed dose to demonstrate how elevated the ingested dose is, anyway.
We added a supplementary file 2 with formula and with an example of calculation.
Comment 4: The “whole body store” standard seems contradicted by the Ambrose, et al. study (reference 21) in which IM thiamine was given at doses of 5, 10, 20, 50, 100, or 200 mg a day for two days: “A planned comparison between the 200 mg group and the mean of the other dosage groups was significant…” (Ambrose et al., 2001, p. 114). As whole-body stores are 20-30 mg, all doses above 20 mg should have been equivalent.
Response 4: We recalculated the statistics of the Ambrose's paper concerning eye signs and ataxia, and we do not agree with the conclusion of the authors: there is no statistical sign of a better improvement in patients receiving higher dose from 5 to 200mg, groups were too small to reach statistical power. The major point is elsewhere: even with very low doses of thiamine, patients were not worst. Since no enough data were provided to allow recalculation, we have to admit the core statistics of this paper concerning better cognitive results in patients receiving higher doses (50mg better than 20mg): however many flaws were not addressed. It is difficult to exclude such a citated paper, but one should be careful concerning the claimed results. We added a cautionary statement: "although the results should be carefully examined" line 182.
We knew the paper from Galvin, which supposed a failure of thiamine absorption based on the literature review made by Hoyumpa. (Galvin is not cited in our main text due to restricted place, but his paper was used in the supplementary excel file tabulating protocols line 28)
Comment 5: Also, the ability to use and store thiamine likely depends on the Michaelis dissociation constant for the associated enzymes, which are certainly altered in enzyme polymorphisms, mutations (Ames, et al., 2002) and aging (Ames, et al., 2006) and might be altered in disease states such as Wernicke’s encephalopathy (Galvin, et al., 2010).
Response 5: We agree that absorption might be modified by chronic alcohol intake, aging or by enzymatic polymorphisms, that's why calculated the corrected values which neutralize the amounts of thiamine given PO and possibly poorly absorbed. Although these corrections mitigated orally given amounts thiamine, a still massive thiamine supplementation remains !
Comment 6: Nonetheless, the author makes a good case when reviewing data for the rapid resolution of lactic acidosis and cardiac symptoms and the relatively flat dose-response curve above I00 mg IM. I think emphasizing this type of data and providing a more nuanced understanding of thiamine stores will strengthen an already good manuscript.
Response 6: We thank the reviewer for this comment. Unfortunately, we failed to found other reports examining the very early response of lactic acidosis to thiamine.
We agree that our approach keeps thiamine biochemistry into a black box and mainly emphasizes the excessive whole amount of thiamine used to supplement deficient patients. We hope that the response 3 have clarified our opinion concerning the whole-body store, which remains a very conservative empirical approach allowing to clearly raise the problem of doses.
Reviewer 2 Report
Comments and Suggestions for Authors
The manuscript titled "Protocols of thiamine supplementation: clinically driven rationality vs biological commonsense" analyzes current protocols for thiamine (vitamin B1) supplementation, particularly in the neurological context. The author compares clinically recommended doses with those biologically required, advocating for a “biological commonsense” approach. However, it would be helpful to more clearly explain the physiological reasoning that justifies the effectiveness of lower doses.
- For example, the manuscript should clarify how saturation of intestinal transporters or the tissue storage capacity of thiamine affects therapeutic efficacy.
- Greater experimental support:
The work is based on a review of protocols and literature, but it lacks direct experimental or clinical validation that systematically compares different thiamine dosages.
It would be useful to propose (or at least suggest) an experimental design that could validate the efficacy of a single 100 mg dose. - More in-depth discussion of clinical scenarios:
The distinction between subclinical and severe forms (such as Wernicke’s syndrome or beriberi) is mentioned but not fully developed.
It should be better explained in which cases a single dose is sufficient, and when a prolonged or combined (IV/oral) regimen may be justified. - Graphical clarity:
Figures (e.g., Fig. 1 and 2) are interesting but require more readable and descriptive legends. Some concepts (such as “fold vs. whole-body store”) should be explained in the main text to make them accessible to non-expert readers. - More balanced conclusions:
The conclusions state that a single dose is probably sufficient, but this position should be better contextualized by emphasizing the need for clinical trials to confirm it definitively.
Author Response
Comment 1: The manuscript titled "Protocols of thiamine supplementation: clinically driven rationality vs biological commonsense" analyzes current protocols for thiamine (vitamin B1) supplementation, particularly in the neurological context. The author compares clinically recommended doses with those biologically required, advocating for a “biological commonsense” approach. However, it would be helpful to more clearly explain the physiological reasoning that justifies the effectiveness of lower doses.
Response 1: We propose to stratify the response to this comment.
*Intestinal transportation rate is debated: Thomson 1972, 1970, 2000, and Chataway, 1995 suggested a limited absorption rate; whereas Yoon 2019 and Weber 1985 did not confirm this limitation. This point is explained in paragraph 3.2.
To completely circumvent this possible limited absorption, we always calculated the corrected doses which corrected oral dose using the lowest absorption rate of the literature.
*The goal of the study is not to justify lower dose, but to observe that high doses are provided.
We have to admit that except papers dedicated to shoshin beriberi which demonstrate that low thiamine completely reverses severe cardiac insufficiency and lactic acidosis, no study examined effects of low dose thiamine in neurology, except those which are cited in references.
*Thiamine doses are considered high in comparison with the whole-body thiamine store (20-30mg). This value is largely available in the literature, and changes in the limits of the values would not really change our results.
*We agree that benfotiamine, a derivative of thiamine, is better absorbed after oral intake. However, this drug is not available in France, moreover it might be excluded from brain (Volvert BMC Pharmacol 2008). So, we preferred not to develop this point.
*We added a reference to a complete review dealing with the complexity of intestinal transporters of thiamine.
Comment 2: For example, the manuscript should clarify how saturation of intestinal transporters or the tissue storage capacity of thiamine affects therapeutic efficacy.
Response 2: To our knowledge, there is no data linking therapeutic efficacy and intestinal transporters or tissue storage, except in rare genetic disorders affecting transporters. It is only commonly admitted that thiamine is poorly absorbed in chronic alcohol intakes (Thomson). We are not aware of any paper sizing the differential tissue storage of thiamine during various disorders. However, an empiric proof of tissue storage deficiency is empirically obtained from deficient intake: example in gravidarum hyperemesis which leads to GW in a few weeks.
Comment 3: Greater experimental support: The work is based on a review of protocols and literature, but it lacks direct experimental or clinical validation that systematically compares different thiamine dosages. It would be useful to propose (or at least suggest) an experimental design that could validate the efficacy of a single 100 mg dose.
Response 3: We agree that experimental validation of different doses is lacking. However, low dose (100mg) is already largely used in Canada (see in ref Day 2015 figure 4). We added a point in the discussion (Line 210).
Comment 3: More in-depth discussion of clinical scenarios: The distinction between subclinical and severe forms (such as Wernicke’s syndrome or beriberi) is mentioned but not fully developed. It should be better explained in which cases a single dose is sufficient, and when a prolonged or combined (IV/oral) regimen may be justified.
Response 3: In our opinion, each clinical subset, Wernicke's syndrome, Korsakoff, Neuropathy, or cardiac beriberi, are beriberi (we added a reference to the classical book of Maurice Victor and Adams). As a rule of thumb, and due to an unknown mechanism, vitamin deficiencies are prone to partial expression: e.g. B12 deficiency provokes neuroBiermer or anemia.
Therefore, each clinical subset of thiamine deficiency should be equally considered to be a deficiency requiring treatment, which might be a simple replenishment of the body store. Depending if you are confident with authors limiting intestinal transportation or not, IV or PO ways might allow almost the same biodisponibility. Frankly, unique 100mg IV thiamine is probably sufficient to overcome deficiency and intestinal transport in all cases, but I can't write it in the paper since studies are lacking.
It is to note that in Victor and Adams in their classical book on Wernicke used 50 to 100mg and observed improvement within hours. Major increase in thiamine doses is therefore a recent tradition.
Comment 4: Graphical clarity: Figures (e.g., Fig. 1 and 2) are interesting but require more readable and descriptive legends. Some concepts (such as “fold vs. whole-body store”) should be explained in the main text to make them accessible to non-expert readers.
Response 4: We explained the concept and we gave an example of interpretation.
We also improved Figure 2.
Comment 5: More balanced conclusions: The conclusions state that a single dose is probably sufficient, but this position should be better contextualized by emphasizing the need for clinical trials to confirm it definitively.
Response 5: We added a sentence in discussion to emphasize the need for clinical trials.
Round 2
Reviewer 1 Report
Comments and Suggestions for Authors
I appreciate the author's thorough and well-reasoned responses to the v1 review.